# Precipitant Effects on Aggregates Structure of Asphaltene and Their Implications for Groundwater Remediation

**Christian B. Hammond [1]**, **Dengjun Wang [2]** and **Lei Wu [1],***

[1]  Department of Civil Engineering, Ohio University, Athens, OH 45701, USA; ch096417@ohio.edu
[2]  Oak Ridge Institute for Science and Education (ORISE), U.S. Environmental Protection Agency (EPA), Ada, OK 74820, USA; wang.dengjun@epa.gov
*  Correspondence: wul@ohio.edu

**Abstract:** Asphaltenes generally aggregate, then precipitate and deposit on the surfaces of environmental media (soil, sediment, aquifer, and aquitard). Previous studies have recognized the importance of asphaltene aggregates on the wettability of aquifer systems, which has long been regarded as a limiting factor that determines the feasibility and remediation efficiency of sites contaminated by heavy oils. However, the mechanisms/factors associated with precipitant effects on asphaltene aggregates structure, and how the precipitant effects influence the wettability of surfaces remain largely unknown. Here, we observe the particle-by-particle growth of asphaltene aggregates formed at different precipitant concentrations. Our results show that aggregates for all precipitant concentrations are highly polydisperse with self-similar structures. A higher precipitant concentration leads to a more compacted aggregates structure, while precipitant concentration near to onset point results in a less compact structure. The well-known Smoluchowski model is inadequate to describe the structural evolutions of asphaltene aggregates, even for aggregation scenarios induced by a precipitant concentration at the onset point where the Smoluchowski model is expected to explain the aggregate size distribution. It is suggested that aggregates with relative high fractal dimensions observed at high precipitant concentrations can be used to explain the relatively low Stokes settling velocities observed for large asphaltene aggregates. In addition, asphaltene aggregates with high fractal dimensions are likely to have high density of nanoscale roughness which could enhance the hydrophobicity of interfaces when they deposit on the sand surface. Findings obtained from this study advance our current understandings on the fate and transport of heavy oil contaminants in the subsurface environment, which will have important implications for designing and implementing more effective and efficient remediation technologies for contaminated sites.

**Keywords:** asphaltene aggregation; fractal aggregates; precipitant; groundwater remediation

## 1. Introduction

Crude oil is a common soil contaminant usually originating from a variety of sources, including petroleum spills and fuel storage tank leakages. Upon its release to the ground surface, crude oil has the propensity to migrate into the subsurface where it can persist as hydrophobic contaminants and occasionally leach to nearby ground and surface waters. Driven by significant adverse environmental impacts and risks, the detection and remediation of crude oil contaminated soils and aquifers have yielded a large amount of research interest in the past decades [1–5]. Solvent extraction is a well-accepted method for the removal of hydrophobic organic contaminants from soil. The Soxhlet extraction technique outlined by the U.S. Environmental Protection Agency (EPA) method 3540C

describes the process and solvents used for crude oil contaminated soil remediation in detail [6]. However, solvent extraction can trigger changes in the solvation characteristics of crude oil potentially undermining the efficiency of the remediation process.

Crude oils contain asphaltenes, which are high-molecular-weight polar compounds that are known to be insoluble in light *n*-alkanes and soluble in aromatic solvents [7,8]. Asphaltenes generally aggregate, and then precipitate and deposit on surfaces of environmental matrixes (e.g., soil, sediment, aquifer, and aquitard). Previous studies have shown significant effects of asphaltenes on the wettability of aquifer systems [9–11]. In addition to the classical mechanisms controlled by Derjaguin–Landau–Verwey–Overbeek (DLVO) interactions between asphaltene particles and sand surface, asphaltene aggregation and precipitation are other important mechanisms that can change the wettability of sand, which can have significant impacts on groundwater remediation [9]. Innovations in surfactant flushing and solvent extraction techniques for crude oil contaminant remediation can cause changes in the solvation characteristics of the crude oil, causing asphaltene aggregation, changing the wettability of subsurface system, and hence altering the distribution and recoverability of these contaminants. Although the importance of reversal of wettability caused by the aggregation of asphaltene has been recognized, fundamental understandings on this process remain elusive [1,9,10]. Using an atomic force microscopy (AFM) imaging technique, Zheng et al. [10] have shown that the asphaltene aggregates can grow up to 6000 nm in diameter and 3500 nm in thickness, which are orders of magnitude greater than the dimensions captured by DLVO theory, indicating that DLVO interactions are not the dominant mode of changing the wettability property on a quartz surface. Moreover, the aforementioned study mainly focused on measuring the size (dimension) of asphaltene aggregates deposited on quartz surfaces, without including any details on aggregate structure analysis. Hence, the effects of changing solvation characteristics of crude oil on the formation of asphaltene aggregates with different structures, and associated effects on wettability of sand surfaces, are not clear. The novelty of this study is to provide a fundamental understanding of how precipitant concentration affects asphaltene aggregates structure, which eventually has important implications on wettability of aquifer surfaces.

The study of asphaltene aggregation and precipitation due to changes in pressure, temperature [12,13] and composition [14] have been well documented at different length scales from molecular to colloidal sizes. The focus of this study is on the investigation of colloidal properties of asphaltene aggregates formed at different precipitant concentrations. The aggregates are defined as clusters of asphaltene particles with sizes up to several micrometers in average diameter. The colloidal asphaltene precipitation is observed when the asphaltene aggregates reach a detectable size as a precipitant (i.e., *n*-heptane) is added to the crude/model oil. Deposition refers to the settling of the asphaltene aggregates onto the solid surface. Existing studies on asphaltene aggregation induced by the addition of precipitant into crude oil have focused on the kinetics of aggregate formation with limited insights on effect of structure [8,15–18]. It is well accepted that the kinetics of asphaltene aggregation strongly depend on the solubility parameters of the mixture of crude oil, solvent, and precipitant. When precipitants are added at concentrations near the precipitation onset point, asphaltene aggregation can be described by a diffusion-limited aggregation (DLA) process [15]. A slight change in precipitant concentration away from precipitation onset point leads to a crossover between diffusion-limited aggregation (DLA) and reaction-limited aggregation (RLA) [15]. The characteristic time of aggregation decreases exponentially upon increase in the precipitant concentration. However, quantitative results regarding asphaltene aggregate structure are not well understood, with different studies even reporting conflicting results. Yudin et al. and Ashoori et al. studied the kinetics of precipitated, extracted, and re-dissolved asphaltenes in toluene. They found an average value of fractal dimension $d_f = 1.7 \pm 0.2$ for DLA [18,19] while Huang et al. and Seifried et al. reported much larger values of $d_f$ for crude oil in the DLA regime [8,15]. Moreover, Hoepfner et al. concluded that no clear trend was observed for the fractal dimensions of asphaltenes aggregates as a function of precipitant (*n*-heptane) concentration [17]. Therefore, a high-resolution experimental study, which allows a direct analysis of aggregate structure,

is strongly needed to better understand the subtle differences in fractal dimensions at different precipitant concentrations.

The overarching objective of this study is to investigate the effects of precipitant concentrations on asphaltene aggregates structure. To determine the sensitivity of aggregates structure to different precipitant concentrations, a wide range of concentrations from close to precipitation onset point and higher were evaluated. A combined technique of in-situ microscopy observation and advanced imaging analysis was used to study aggregate morphological properties under different precipitant concentrations. To reduce the effects of complex fractions typically found in crude oil, the current study used a model oil, prepared by dissolving extracted asphaltene in toluene.

## 2. Experiments

### 2.1. Materials

Model oil used in this project was prepared from crude oil from Venezuela with an asphaltene content of 9.0 wt%. The results of elemental analysis of extracted asphaltenes show a total heteroatom (nitrogen, sulfur, and oxygen) content of approximately 10 wt%. Toluene (>99%, Thermo Fisher Scientific Inc., Waltham, MA, USA) and *n*-heptane (>95%, Thermo Fisher Scientific Inc., Waltham, MA, USA) were used as the solvent and precipitant, respectively.

To extract asphaltenes, the crude oil was mixed with *n*-heptane in a 1:40 ratio and stirred for 24 h. The precipitated asphaltene solids were separated from the oil/heptane mixture by centrifugation. The collected asphaltene solids were washed with *n*-heptane for 24 h to remove any non-asphaltene material. The asphaltenes were then dried in an oven at 70 °C. To prepare the model oil, dried asphaltenes were dissolved in toluene to a target concentration of 1 wt% and stirred continuously for 5 days to fully dissolve the asphaltene. The details of asphaltene extraction and model oil preparation were given elsewhere [17].

### 2.2. Experiments

Prior to the aggregation study, preliminary experiments were conducted to determine the instantaneous onset point of precipitation of asphaltene which is defined as the concentration of *n*-heptane (vol%) $\varphi$ that has to be added to the model oil (i.e., asphaltene/toluene) to observe insoluble asphaltenes via optical microscopy in less than 10 min of mixing with the precipitant. In this study, the instantaneous onset concentration for the prepared model oil was measured at 52 vol% *n*-heptane, which is in agreement with reported onset concentrations of precipitation for asphaltene [18]. To test the sensitivity of asphaltene aggregate structure upon changing the precipitant concentration, three experiments were carried out: (1) adding 52 vol% of *n*-heptane to initiate the aggregation, where $\varphi \approx \varphi_{onset}$, (2) adding 55 vol% of *n*-heptane to initiate the aggregation, where $\varphi > \varphi_{onset}$, and (3) adding 57 vol% of *n*-heptane to initiate the aggregation, where $\varphi \gg \varphi_{onset}$.

The experiments of aggregation were conducted using a liquid cell mounted on a 100× magnification inverted optical microscope (Eclipse Ti-E, Nikon Instruments Inc., Melville, NY, USA). A $3 \times 3$ large image with a normal field of view was taken at the end of each experiment and stitched together to obtain larger statistic of aggregates for structural analyses of mature aggregates under steady-state conditions. The details of the experimental set up and procedures were described previously [20,21].

### 2.3. Image Analysis

All analyses were performed in MATLAB (The MathWorks, Inc, Natick, MA, USA) and the key steps are summarized here. First, a global threshold via Ostwald criterion was used to generate binary images from bright-field images (Figure 1). Then, all objects were identified, labeled, and characterized by segmentation, using *bwlabel* and *regionprops* functions in MATLAB. Identified objects were also fitted with best-fit ellipses by customized functions. To differentiate

individual particles, small clusters, and large clusters, four morphology parameters were used: area, solidity, solidity_ellipse, and eccentricity. More details of the definition and acceptable criteria of the morphological parameters used in this study are illustrated in Figure 2.

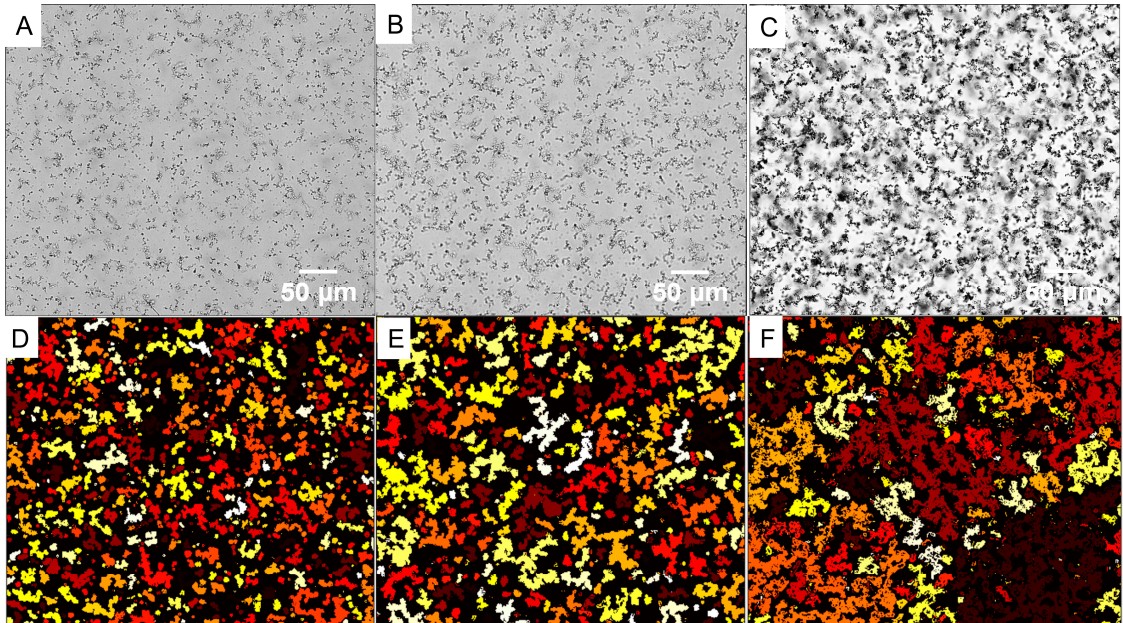

**Figure 1.** Large-stitched images of asphaltene aggregates at the end of each experiment. (**A–C**) represent bright-field images of aggregates formed by adding 52 vol%, 55 vol%, and 57 vol% heptane to model oil, respectively. (**D–F**) represent the corresponding binary images after applying Ostwald's global thresholding criterion. The scale bar is 50 µm.

The fractal dimensions of aggregates were determined by relating the cluster sizes to the radius of gyration of the clusters [17,20]:

$$A = R_g{}^{d_f} \tag{1}$$

where $A$ is the area of the cluster and $R_g$ is the radius of gyration. The radius of gyration can be determined by:

$$R_g = \sqrt{\frac{(x - x_c)^2 + (y - y_c)^2}{N - 1}} \tag{2}$$

where $x_c$ and $y_c$ are the $x$–$y$ coordinates of the centroid of each cluster, $x$ and $y$ are the $x$–$y$ coordinates of each pixel in a cluster, and $N$ is the number of pixels in each cluster.

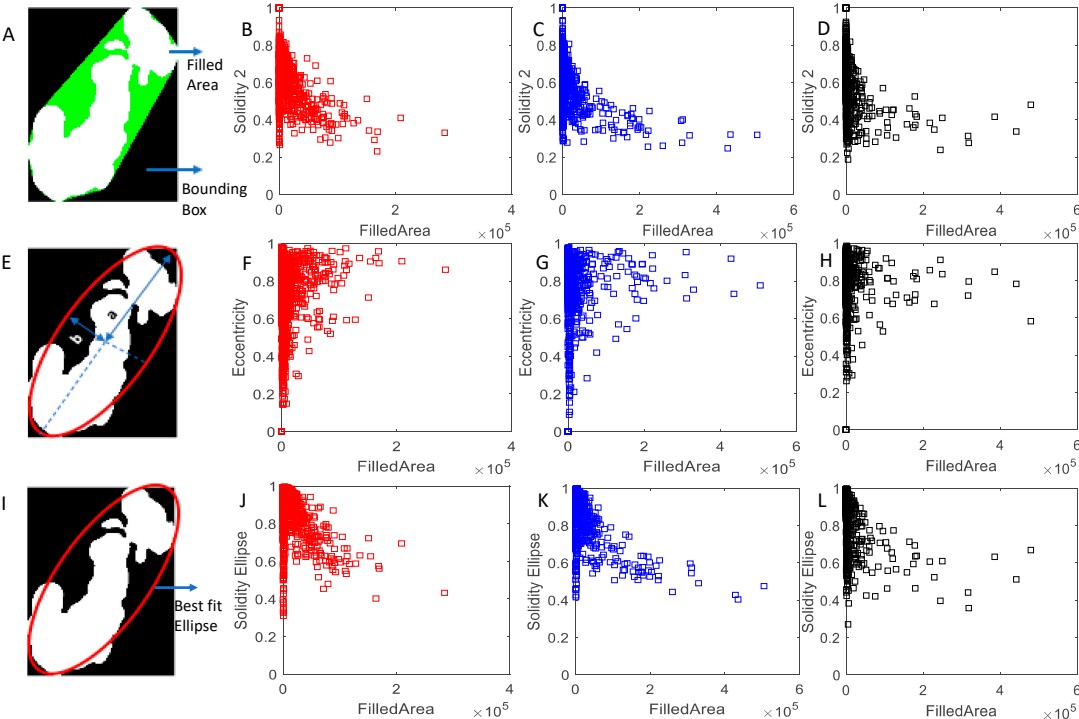

**Figure 2.** Morphology parameters and criteria used for identification of single particle, small and large clusters. (**A,E,I**) demonstrate the definition of Solidity 2, Eccentricity, and Solidity Ellipse, respectively. Specifically, Solidity 2 is the ratio of filled area to bounding box; *Eccentricity* $= \sqrt{1 - (b^2/a^2)}$, where $a$ and $b$ are the semi-major and semi-minor axes of the best fit ellipse respectively; Solidity ellipse is the ratio of filled area to area of the best fit ellipse. (**B–D,F–H,J–L**) represent plots of Solidity 2, Eccentricity, and Solidity Ellipse as a function of Filled Area (in pixel) of aggregates formed respectively, upon addition of 52 vol% (red squares), 55 vol% (blue squares), and 57 vol% (black squares) *n*-heptane. Small clusters were identified under the following criterion: Solidity 2 < 0.9 & Solidity 2 > 0.7 & Eccentricity < 0.5. Large clusters were identified under the following criterion: Eccentricity > 0.7 or Solidity_ellipse < 0.9. Single particles were identified under the following criterion: Solidity 2 < 0.9 & Solidity 2 > 0.7 & Eccentricity < 0.5 & Filled Area < 2000 (in pixel).

## 3. Results and Discussion

Figure 1 shows representative structure images of asphaltene aggregates formed at different precipitant concentrations. We found that these aggregates have rugged, irregular structures and show fractal characteristics at all precipitant concentrations. To further elucidate the fractal characteristics of aggregates, the aggregate fractal dimensions ($d_f$) were computed by the following scaling law using the area and the radius of gyration: $A \propto R_g{}^{d_f}$ (Figure 3). More details on the goodness of fit are shown in Table 1. For asphaltene aggregates formed at different precipitant concentrations, the sizes of the aggregates $R_g$ were in a wide range of 10–1000 μm, suggesting these aggregates are highly polydisperse. The higher precipitant concentration resulted in a slight increase of initial particle size from 1 to 4 μm, indicating a faster aggregation rate in a short time. The aggregates for all three experiments described above have a fractal scaling region that spanned over two orders of magnitude in length. However, there is a change in the scaling behavior at $R_g$ = 30 μm for all precipitant concentrations, suggesting that the structure of fractal clusters changes as they transition from small to large clusters. For small clusters, regardless of the precipitant concentration, the fractal dimensions of asphaltene aggregates are nearly identical (≈2), indicating a more compact and dense structure. The compact structure of small clusters observed in this study is consistent with previously reported results [17]. The compact structure observed for small clusters can be attributed to cluster restructuring often seen at smaller length scales [17]. Small aggregate clusters were also observed

at a precipitant concentration of 57 vol%, which could imply that some asphaltenes are very stable which can only form small fractal aggregates even at high precipitant concentrations. For large clusters, the fractal dimensions were determined to be 1.66 ± 0.17, 1.73 ± 0.17, and 1.83 ± 0.28 for precipitant concentrations at 52 vol%, 55 vol%, and 57 vol%, respectively. A slightly higher fractal dimension with increasing precipitant concentrations indicates that aggregates are less amorphous with more compacted structures. Approaching the precipitation onset point (52 vol% and 55 vol%), the fractal dimensions determined in this study are in good agreement with the findings from the DLA in classic colloids and previous asphaltene aggregation studies [17,22]. Although the mechanisms/factors associated with the observation of compact aggregates structure at higher precipitant concentration are not immediately clear, it is likely due to increased interaggregate attractions (e.g., London dispersion force) or the removal of a steric stabilization barrier [17]. Nevertheless, the fractal dimensions based on image analysis used in this study fall in the range of the $d_f$ values reported in the literature for asphaltene aggregates [15,17]. It is worth noting that the large stitched images used our analysis are two-dimensional projected areas of actual three-dimensional aggregates. Thus, some of the details of the original aggregate structure were obscured in the projection of the resultant profile. The $d_f$ determined here is two-dimensional and therefore intrinsically smaller than $d_f$ determined from three-dimensional based measurement such as light scattering or settling velocity. Thus, the differences between the $d_f$ measured by image analysis and the $d_f$ determined based on the latter techniques should be taken into consideration.

We also determined the cluster size distribution for all experiments as depicted in Figure 4. The results show that the data are skewed to the left of the mean for all precipitant concentrations, indicating the aggregate growth is not only dominated by diffusion-limited cluster-cluster aggregation in the later stage, but some non-classical growth mechanisms such as cluster-particle collision and coalescence instead. The results for precipitant concentrations at/near the onset point (experiments 1 and 2) are very similar but different at higher precipitant concentrations (experiment 3). The deviations are most significant for the tail of the distribution, where the data suggest that aggregates formed at the higher precipitant concentration (57 vol%) have a heavier-tailed distribution than those near the precipitant onset point (52 and 55 vol%). The heavy tail distribution at the higher precipitant concentration results in a larger mean aggregate size, which is in good agreement with previous studies [23]. We also compared the cluster size distribution to the Smoluchowski kinetics, that describe the time evolution of an ensemble of particles as they aggregate. The details of Smoluchowski model can be found elsewhere [24]. The results show that Smoluchowski model cannot describe the observations captured in this study, even for aggregation induced by precipitant at the onset point, where the Smoluchowski model is expected to describe cluster size distribution. Specifically, the model underestimates the number of clusters smaller than the mean at precipitant concentrations near the onset point and overestimates the number of clusters that are larger than the mean. These discrepancies indicate that the Smoluchowski model is insufficient to explain the aggregate size distribution. Thus, to fully reveal the mechanisms governing asphaltene aggregation at different precipitant concentrations, both aggregation kinetics and aggregate structure must be examined.

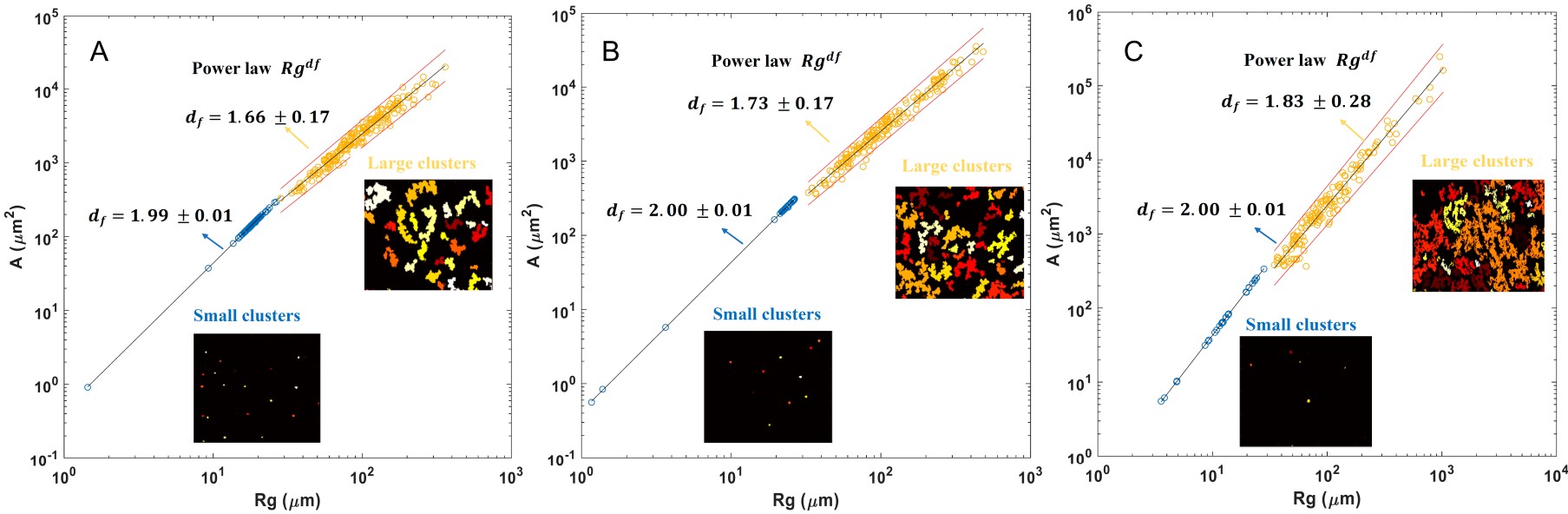

**Figure 3.** Log-log plots of average area *A* as a function of the radius of gyration $R_g$ for asphaltene aggregates formed under different precipitant concentrations. (**A–C**) represents aggregation induced by adding 52 vol%, 55 vol%, and 57 vol% heptane to model oil, respectively. The solid dark blue and black lines represent the best fit line for small clusters and large clusters, respectively. The solid red lines represent the prediction bounds for the fitted functions and the bounds reflect 95% confidence intervals. Single particles were excluded from the regression.

**Table 1.** Estimation of fractal dimension by least square fitting for power law of $R_g{}^{d_f}$.

| Experiments | Cluster | Goodness of Fit | | | | | |
|---|---|---|---|---|---|---|---|
| | | Best fit $d_f$ | $R^2$ | SSE | DFE | Adjusted $R^2$ | RMSE |
| 1 | Large | 1.66 ± 0.17 | 0.96 | 5.77 | 203 | 0.96 | 0.17 |
| | Small | 1.99 ± 0.01 | 0.99 | 0.009 | 82 | 0.99 | 0.01 |
| 2 | Large | 1.73 ± 0.17 | 0.97 | 4.88 | 163 | 0.97 | 0.17 |
| | Small | 2.00 ± 0.01 | 0.99 | 0.006 | 34 | 0.99 | 0.013 |
| 3 | Large | 1.83 ± 0.28 | 0.96 | 10.87 | 138 | 0.96 | 0.28 |
| | Small | 2.00 ± 0.01 | 0.99 | 0.004 | 28 | 0.99 | 0.012 |

Note: Experiments 1, 2, and 3 represent asphaltene aggregation induced by adding precipitant (heptane) at 52 vol%, 55 vol% and 57 vol%, respectively. SSE: sum of squared estimate of errors; DFE: degrees of freedom for error; $R^2$: R-squared; and RMSE: root mean square error.

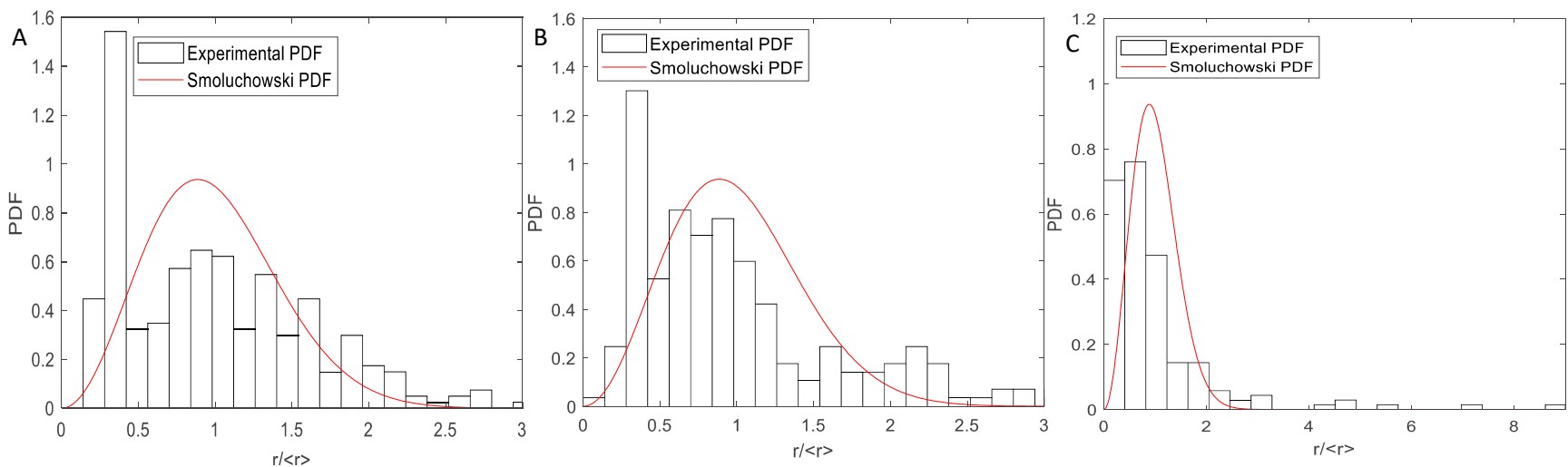

**Figure 4.** Cluster size distribution of aggregates formed at different precipitant concentrations. (**A–C**) represents aggregation induced by adding 52 vol%, 55 vol%, and 57 vol% heptane into model oil, respectively. Cluster size distributions for each experiment are normalized by their respective mean values to facilitate direct comparison. The red lines are the particle size distribution for Smoluchowski kinetics.

## 4. Conclusions and Environmental Implications

We observed asphaltene aggregates formation at different precipitant concentrations using in situ observations combined with advanced statistical analyses. Our results show that aggregates for all precipitant concentrations are highly polydisperse with self-similar structures. Higher precipitant concentration leads to a more compacted aggregates structure, while precipitant near to onset point results in a less compact structure. The Smoluchowski model is inadequate to describe the structural evolutions of asphaltene aggregates, even for aggregation scenarios induced by precipitant at the onset point where Smoluchowski model is expected to explain the aggregate size distribution. The fractal dimension values reported here furnish important insights on the permeability and compactness of the asphaltene aggregates, which provides the fundamentals for a more accurate prediction of critical parameters (e.g., particle transport and settling velocities, and wettability of porous media) that determine the feasibility and efficiency of in situ contaminated site remediation. For instance, aggregates with relative high fractal dimensions observed at high precipitant concentrations can be used to explain the relatively low Stokes settling velocities observed for large asphaltene aggregates (Figure 5). This can be attributed to internal flow in the aggregates that does not appreciably reduce the drag force exerted by the surrounding liquid (fluid). The effect of the internal flow on the aggregates depends on permeability and pore size [25]. For aggregates formed at high precipitant concentrations, the growth in the later stage is dominated by particle-cluster collisions, which result in denser (small pore) structures in the inner part of aggregate, and less dense (large pore) structures on the outside. Also, the size and morphology of asphaltene aggregate deposits have non-trivial effects on wettability of aquifer systems (porous media). It is suggested that asphaltene aggregates with different fractal dimensions observed in this study have different nanoscale morphological parameters such as geometry and roughness, which could significantly alter the surface energy and thereby the wettability of grain/sand surfaces (Figure 5) [26]. More specifically, asphaltene aggregates with high fractal dimension are likely to have high density of nanoscale roughness which could enhance the hydrophobicity of interfaces when they deposit on surfaces. These results have important practical implications, especially within the big framework on how to distribute and recover heavy oils in contaminated soils and aquifers. This is especially useful in the application of solvent flushing-based remediation technologies (e.g., solvent blends of acetone-hexane) [27] to dissolve or mobilize heavy soil in the subsurface. The effectiveness of solvent extraction remediation technologies depends on firm contact between the soil surface and the solvents. Typically, the idea behind solvent extraction is the ability of solvents to dissolve oil contaminants for easier mobilization and recovery. However, an undesirable side effect could result if the solvation power of the heavy oil is reduced and the wettability of the surface is altered due to the asphaltene aggregates deposits. The findings from our study shed light on some unresolved issues in remediation technology. For instance, Li et al. examined the effectiveness of acetone-hexane solvent blends for heavy oil contaminated soil remediation and found that among the different chemical fractions in crude oil, the removal efficiency of asphaltene fraction was the lowest [27]. Perhaps, this low removal efficiency could be attributed to asphaltene aggregate formation and deposition, triggered by the application of solvent-blends. The results from our study highlight the need for careful consideration of solvent types/blends and solvent blend ratios used for solvent extraction remediation of crude oil contaminated soils. Therefore, a more thorough understanding of the effects of precipitant concentration on asphaltene aggregate structure and associated effects on its fate and transport in the subsurface environment is needed for designing more effective and efficient heavy oil remediation technologies. Future research efforts should include high resolution studies that allow for the digitization of the settlement trajectory of asphaltene aggregates with different structure in porous media to calculate settling velocity and directly measure nanostructure, surface topography, and contact angle of the asphaltene aggregates and sand interfaces to determine the wettability property.

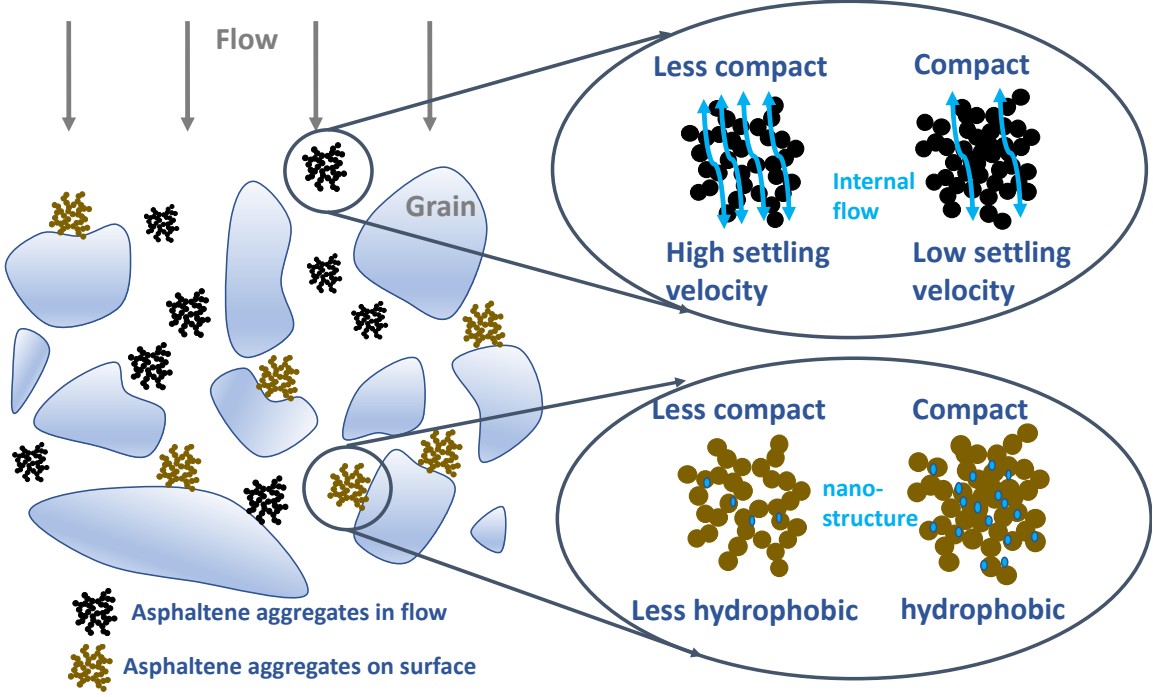

**Figure 5.** Schematic of the proposed asphaltene aggregates structure effects on settling velocity and wettability of aquifer surface, and associated implications for in situ remediation of sites contaminated with heavy oils.

**Author Contributions:** Conceptualization, L.W.; Data curation, C.B.H.; Formal analysis, C.B.H.; Methodology, L.W.; Project administration, L.W.; Writing—original draft, C.B.H.; Writing—review & editing, D.W. and L.W. All authors have read and agreed to the published version of the manuscript.

**Funding:** This work was supported by the following grants to L.W.: the start-up funds from Ohio University and the US National Science Foundation (NSF) #1836905.

**Conflicts of Interest:** The authors declare no conflict of interest.

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
