# Peer review of "Precipitant Effects on Aggregates Structure of Asphaltene and Their Implications for Groundwater Remediation"

_water, doi:10.3390/w12082116_

Round 1

Reviewer 1 Report

Good description and approach to a relevant environmental problem. Although relatively general, and requiring further studies and developments, this manuscript gives already a very sound insight to the scientific issues behind the described problem.

Author Response

REVIEWER # 1

RC: Good description and approach to a relevant environmental problem. Although relatively general, and requiring further studies and developments, this manuscript gives already a very sound insight to the scientific issues behind the described problem.

AR: Thanks for your encouraging comment. We addressed all the comments and suggestions in the revised manuscript.

Reviewer 2 Report

1- I suggest the authors add the conclusion section to the manuscript.

2- I strongly recommend the authors to add one paragraph discussing the difference between their work and the previously performed studies in the literature. In other words, what is the novelty of this work? I offer the authors to revise the abstract and introduction in order to incorporate the novelty of their work. 

3- The authors should have the manuscript edited by a native English speaker in order to help clarify the major conclusions of the article.

Author Response

REVIEWER # 2

RC1:  I suggest the authors add the conclusion section to the manuscript.

AR: Thanks for this suggestion. We agree. We have added some concluding remarks generated from the experiments and merged the conclusion section with the environmental implication section to generate the new “Conclusions and Environmental Implications” section in the revised manuscript.

RC2:  I strongly recommend the authors to add one paragraph discussing the difference between their work and the previously performed studies in the literature. In other words, what is the novelty of this work? I offer the authors to revise the abstract and introduction in order to incorporate the novelty of their work. 

AR: Thanks for your insights. We summarized the key findings of prior studies in the literature, pointed out the associated knowledge gap and derived the motivation (novelty) of our work in the second paragraph of the manuscript. Highlighted in the revised manuscript. Specifically, existing studies generally focused on using the classical DLVO interactions between the asphaltene particles and sand surfaces to explain the reversal of surface wettability. However, DLVO mechanism is not the only dominant mechanism controlling the aggregation behaviors observed in our study. We therefore explored additional factors and mechanisms on the possible influence of aggregate structure on changing the wettability of aquifer surfaces in our study. The outcome of this work will add new knowledge on particle aggregation in aquatic environments, which opens the door for better designing and implementing effective and efficient strategies for many practical applications like contaminated site remediation.

RC3: The authors should have the manuscript edited by a native English speaker in order to help clarify the major conclusions of the article.

AR: Thanks. We have reviewed the article again and have employed the services of a native English to review thoroughly as well. This is addressed in the revised manuscript.

Round 2

Reviewer 2 Report

The authors have satisfactorily responded to all questions and made the necessary changes to the manuscript; however, in the section about the challenges in asphaltene aggregation; they also need to mention the differences between asphaltene aggregation, precipitation and deposition. They can use the following articles as the reference:

1) Cruz, Arley A., et al. "CO2 influence on asphaltene precipitation." The Journal of Supercritical Fluids 143 (2019): 24-31.

2) Dashti H, Zanganeh P, Ayatollahi S. The comparison between heavy and light oil asphaltene deposition during pressure depletion and CO2 injection at reservoir condition, a visual laboratory study. Chemeca 2013: Challenging Tomorrow. 2013:833.

3) Jia, Bao, Jyun-Syung Tsau, and Reza Barati. "A review of the current progress of CO2 injection EOR and carbon storage in shale oil reservoirs." Fuel 236 (2019): 404-427.

4) Kord, Shahin, et al. "Evaluation of the Kinetics of Asphaltene Flocculation during Natural Depletion and CO 2 Injection in Heptane-Toluene Mixtures." SPE/IATMI Asia Pacific Oil & Gas Conference and Exhibition. Society of Petroleum Engineers, 2017.

5) Dashti, Hossein, et al. "Mechanistic study to investigate the effects of different gas injection scenarios on the rate of asphaltene deposition: An experimental approach." Fuel 262 (2020): 116615.

6) Zanganeh, Peyman, et al. "Asphaltene deposition during CO2 injection and pressure depletion: a visual study." Energy & fuels 26.2 (2012): 1412-1419.

7) Duran, J. A., et al. "Nature of asphaltene aggregates." Energy & Fuels 33.5 (2018): 3694-3710.

8) Vilas Bôas Fávero, Cláudio, et al. "Mechanistic investigation of asphaltene deposition." Energy & Fuels 30.11 (2016): 8915-8921.

Author Response

The comments of the reviewer are very much appreciated and have helped improve the manuscript significantly. We responded to all the comments and revised manuscript accordingly. Below we explain in detail how we responded to each of the comments (RC: Reviewer comment; AR: Author response).

REVIEWER

RC: The authors have satisfactorily responded to all questions and made the necessary changes to the manuscript; however, in the section about the challenges in asphaltene aggregation; they also need to mention the differences between asphaltene aggregation, precipitation and deposition.

AR: Thanks for this comment. We have clarified the differences between asphaltene aggregation, precipitation and deposition in our study in the revised manuscript. See the third paragraph on page 2 “…The study of asphaltene aggregation and precipitation due to changes in pressure, temperature [12, 13] and composition [14] have been well documented at different length scales from molecular to colloidal sizes. The focus of the current study is on the investigation of colloidal properties of asphaltene aggregates formed at different precipitant concentrations. The aggregates are defined as clusters of asphaltene particles with sizes up to several micrometers in average diameter. The colloidal asphaltene precipitation is achieved when the asphaltene aggregates reach a detectable size as the precipitant (i.e. n-heptane) is added to the crude/model oil. The deposition refers to the settling of the asphaltene aggregates onto the solid surface…”

References:

  1. Cruz, Arley A., Monique Amaral, Denisson Santos, André Palma, Elton Franceschi, Gustavo R. Borges, João A. P. Coutinho, Julio Palácio, and Cláudio Dariva. “CO2 Influence on Asphaltene Precipitation.” The Journal of Supercritical Fluids 143 (2019): 24–31.
  2. Dashti, Hossein, Peyman Zanganeh, Shahin Kord, Shahab Ayatollahi, and Amirpiran Amiri. “Mechanistic Study to Investigate the Effects of Different Gas Injection Scenarios on the Rate of Asphaltene Deposition: An Experimental Approach.” Fuel 262 (2020): 116615.
  3. Duran, J. A., Y. A. Casas, Li Xiang, Ling Zhang, Hongbo Zeng, and H. W. Yarranton. “Nature of Asphaltene Aggregates.” Energy & Fuels 33, no. 5 (2019): 3694–3710.

This manuscript is a resubmission of an earlier submission. The following is a list of the peer review reports and author responses from that submission.